# A Maximum-Entropy Approach to Off-Policy Evaluation in Average-Reward MDPs

Nevena Lazić*    Dong Yin*    Mehrdad Farajtabar*    Nir Levine*    Dilan Görür*

Chris Harris†              Dale Schuurmans†

## Abstract

This work focuses on off-policy evaluation (OPE) with function approximation in infinite-horizon undiscounted Markov decision processes (MDPs). For MDPs that are ergodic and linear (i.e. where rewards and dynamics are linear in some known features), we provide the first finite-sample OPE error bound for a model-based approach, extending existing results beyond the episodic and discounted cases. In a more general setting, when the feature dynamics are approximately linear and for arbitrary rewards, we propose a new heuristic approach for estimating stationary distributions with function approximation. Namely, we formulate this problem as finding the maximum-entropy distribution subject to matching feature expectations under empirical dynamics. We show that this results in an exponential-family distribution whose sufficient statistics are the features, paralleling maximum-entropy approaches in supervised learning. We demonstrate the effectiveness of the proposed OPE approaches in multiple environments.

## 1   Introduction

Recently, there have been considerable advances in reinforcement learning (RL), with algorithms achieving impressive performance on game playing and simple robotic tasks. Successful approaches typically learn through direct (online) interaction with the environment. However, in many real applications, access to the environment is limited to a fixed dataset, due to considerations of cost, safety, or time. One key challenge in this setting is off-policy evaluation (OPE): the task of evaluating the performance of a target policy given samples collected by a behavior policy.

The focus of our work is OPE in infinite-horizon undiscounted MDPs, which capture long-horizon tasks such as game playing, routing, and the control of physical systems. Most recent state-of-the-art OPE methods for this setting estimate the ratios of stationary distributions of the target and behavior policy [Liu et al., 2018, Nachum et al., 2019a, Wen et al., 2020, Nachum and Dai, 2020]. These approaches typically produce estimators that are consistent, but have no finite-sample guarantees, and even the existing guarantees may not hold with function approximation. One exception is the recent work of Duan and Wang [2020], which relies on linear function approximation. They assume that the MDP is linear (i.e. that rewards and dynamics are linear in some known feature space) and analyze OPE in episodic and discounted MDPs when given a fixed dataset of i.i.d. trajectories. They establish a finite-sample instance-dependent error upper bound for regression-based fitted Q-iteration (FQI), and a nearly-matching minimax lower bound.

Our work extends the results of Duan and Wang [2020] to the setting of undiscounted ergodic linear MDPs and non-i.i.d. data (coming from a single trajectory). We provide the first finite-sample OPE

error bound for this case; our bound scales similarly to that of Duan and Wang [2020], but depends on the MDP mixing time rather than horizon or discount. We are not aware of any similar results for off-policy evaluation in average-reward MDPs. Indeed, while OPE with linear function approximation has been well-studied for discounted MDPs [Geist and Scherrer, 2014, Dann et al., 2014, Yu, 2010a], in the undiscounted setting even showing convergence of standard methods presents some difficulties (see the discussion in Yu [2010b] for more details).

Beyond linear MDPs, we consider MDPs in which rewards are non-linear, while the state-action dynamics are still (approximately) linear in some features. Here we propose a novel approach for estimating stationary distributions with function approximation: we maximize the distribution entropy subject to matching feature expectations under the empirical dynamics. Interestingly, this results in an exponential family distribution whose sufficient statistics are the features, paralleling the well-known maximum entropy approach to supervised learning [Jaakkola et al., 2000]. We demonstrate the effectiveness of our proposed OPE approaches in multiple environments.

## 2 Preliminaries

**Problem definition.** We are interested in learning from batch data in infinite-horizon ergodic Markov decision processes (MDPs). An MDP is a tuple $(\mathcal{S}, \mathcal{A}, r, P)$, where $\mathcal{S}$ is the state space, $\mathcal{A}$ is the action space, $r : \mathcal{S} \times \mathcal{A} \to \mathbb{R}$ is the reward function, and $P : \mathcal{S} \times \mathcal{A} \to \Delta_{\mathcal{S}}$ is the transition probability function. For ease of exposition, we will assume that states and actions are discrete, but similar ideas apply to continuous state and action spaces. A policy $\pi : \mathcal{S} \to \Delta_{\mathcal{A}}$ is a mapping from a state to a distribution over actions. We will use $\Pi_\pi$ to denote the transition kernel from a state-action pair $(s, a)$ to the next pair $(s', a')$ under $\pi$. In an ergodic MDP, every policy induces a single recurrent class of states, i.e. any state can be reached from any other state. The expected average reward of a policy is defined as

$$J_\pi = \lim_{T \to \infty} \mathbf{E}\left[\frac{1}{T}\sum_{t=1}^{T} r(s_t, a_t)\right] \quad \text{where } s_{t+1} \sim P(\cdot|s_t, a_t) \text{ and } a_t \sim \pi(\cdot|s_t).$$

Assume we are given a trajectory of $T$ transitions $\mathcal{D}_\beta = \{(s_t, a_t, r_t)\}_{t=1}^{T+1}$ generated by a behavior policy $\beta$ in an unknown MDP. The *off-policy evaluation* problem is the task of estimating $J_\pi$ for a target policy $\pi$.

**Stationary distributions.** Let $\mu_\pi(s)$ be the stationary state distribution of a policy $\pi$, and let $d_\pi(s, a) = \mu_\pi(s)\pi(a|s)$ be the stationary state-action distribution. These distributions satisfy the flow constraint

$$\mu_\pi(s') = \sum_{a'} d_\pi(s', a') = \sum_{s,a} d_\pi(s, a) P(s'|s, a). \tag{1}$$

The expected average reward can equivalently be written as $J_\pi = \mathbf{E}_{(s,a) \sim d_\pi}[r(s, a)]$. Thus, one approach to learning in MDPs from batch data involves estimating or optimizing $d_\pi$ subject to (1). In particular, given data sampled from $d_\beta$ and distribution estimates $\widehat{d}_\pi$ and $\widehat{d}_\beta$, we can estimate $J_\pi$ as $\widehat{J}_\pi = \frac{1}{T}\sum_{t=1}^{T} \frac{\widehat{d}_\pi(s_t, a_t)}{\widehat{d}_\beta(s_t, a_t)} r_t$, as proposed by Liu et al. [2018].

**Linear MDPs.** When the state-action space is large or continuous-valued, a common approach to evaluating or optimizing a policy is to use function approximation. Define the conditional transition operator $\mathcal{P}^\pi$ of a policy $\pi$ as

$$\mathcal{P}^\pi f(s, a) := \mathbf{E}_{s' \sim P(\cdot|s,a), a' \sim \pi(\cdot|s')}[f(s', a')|s, a]. \tag{2}$$

With function approximation, it is convenient to assume that for any policy, $\mathcal{P}^\pi$ operates within a particular function class $\mathcal{F}$, i.e. for any $f \in \mathcal{F}$, $\mathcal{P}^\pi f \in \mathcal{F}$ [Duan and Wang, 2020]. We will assume that $\mathcal{F}$ is the set of functions linear in some (known or pre-learned) features $\phi(s, a) \in \mathbb{R}^m$, such that for some matrix $M_\pi \in \mathbb{R}^{m \times m}$,

$$\mathcal{P}^\pi \phi(s, a) = \sum_{s', a'} \pi(a'|s') P(s'|s, a) \phi(s', a')^\top = \phi(s, a)^\top M_\pi + b_\pi^\top. \tag{3}$$

Note that, unlike existing work, we specifically include a bias term $b_\pi$ in the above model. When $\phi(s, a)$ is a binary indicator vector for $(s, a)$, $M_\pi$ corresponds to the state-action transition matrix

and $b_\pi = 0$. However, $b_\pi$ is non-zero in other settings, such as MDPs with linear-Gaussian dynamics. Similarly to Duan and Wang [2020], we will assume that rewards $r(s, a)$ are linear in the same features: $r(s, a) = \phi(s, a)^\top w$. This assumption will be required for the purpose of analysis.

## 3 Off-policy evaluation

### 3.1 Maximum-entropy stationary distribution estimation

Given a policy $\pi(a|s)$, in order to compute an off-policy estimate of $J_\pi$, we only need to estimate the stationary state distribution $\mu_\pi(s)$. We formulate this as a maximum-entropy problem subject to matching feature expectations:

$$\min_{\mu \in \Delta_S} \sum_s \mu(s) \ln \mu(s) \tag{4}$$

$$\text{s.t.} \sum_{s',a'} \mu_\pi(s')\pi(a'|s')\phi(s',a') = \sum_{s,a} \mu_\pi(s)\pi(a|s) \sum_{s',a'} P(s'|s,a)\pi(a'|s')\phi(s',a'). \tag{5}$$

Note that we have relaxed the original flow constraint (1) over all state-action pairs to only require feature expectations to match, similarly to the maximum-entropy principle for supervised learning [Jaakkola et al., 2000]. Furthermore, under the linear MDP assumption and given the model parameters $(M_\pi, b_\pi)$, the feature expectation constraint can be written as

$$\sum_s \mu_\pi(s)\phi(s,\pi)^\top (I - M_\pi) = b_\pi^\top, \tag{6}$$

where $\phi(s,\pi) = \sum_a \pi(a|s)\phi(s,a)$ are feature expectations under the policy. In Appendix A, we show that the optimal solution is an exponential-family distribution of the following form:

$$\mu_\pi(s|\theta_\pi, M_\pi) = \exp\left(\phi(s,\pi)^\top (I - M_\pi)\theta_\pi - F(\theta_\pi|M_\pi)\right) \tag{7}$$

where $F(\theta_\pi|M_\pi)$ is the log-partition function. The parameters $\theta_\pi$ are the solution of the dual problem:

$$\theta_\pi = \arg\min_\theta D(\theta) := F(\theta|M_\pi) - \theta^\top b_\pi. \tag{8}$$

Note that the dual is convex, due to the convexity of the log-partition function in exponential families. Given a batch of data, we estimate the stationary distribution $\mu_\pi$ by first estimating $\widehat{M_\pi}$ and $\hat{b}_\pi$ using linear regression (see (11)), and then computing a parameter estimate as $\widehat{\theta}_\pi = \arg\min_\theta F(\theta|\widehat{M_\pi}) - \hat{b}_\pi^\top\theta$. When the log-partition function $F(\theta|\widehat{M_\pi})$ is intractable, we can optimize the dual using stochastic gradient descent. Noting that $\nabla_\theta F(\theta|M_\pi) = \mathbf{E}_{\mu_\pi}[(I - M_\pi^\top)\phi(s,\pi)]$, we can obtain an (almost) unbiased gradient estimate using importance weights:

$$\widehat{\nabla}_\theta F(\theta|\widehat{M_\pi}) \propto \sum_{s \in \mathcal{D}_\beta} \frac{\hat{\mu}_\pi(s|\theta, \widehat{M_\pi})}{\hat{\mu}_\beta(s|\widehat{\theta}_\beta, \widehat{M_\beta})}(I - \widehat{M_\pi^\top})\phi(s,\pi) \tag{9}$$

where $\hat{\mu}_\beta(s|\widehat{\theta}_\beta, \widehat{M_\beta})$ is an estimate of the stationary distribution of the behavior policy computed using the same approach (we assume that the behavior policy is known and otherwise estimate it from the data). Finally, we evaluate the policy as

$$\widehat{J}_\pi = \sum_{t=1}^T \rho_t r_t \quad \text{where } \rho_t = \frac{\hat{\mu}_\pi(s_t)\pi(a_t|s_t)}{\hat{\mu}_\beta(s_t)\beta(a_t|s_t)}.$$

In practice, it may be beneficial to normalize the distribution weights $\rho_t$ to sum to 1, known as weighted importance sampling [Rubinstein, 1981, Koller and Friedman, 2009, Mahmood et al., 2014]. This results in an estimate that is biased but consistent, and often of much lower variance; the same technique can be applied to the gradient weights following Chen and Luss [2018]. When the log-normalizing constant is intractable, we can normalize the distributions empirically.

**Linear rewards.** When the rewards are linear in the features, $r(s,a) = \phi(s,a)^\top w$, and $b_\pi \neq \mathbf{0}$, there is a faster way to estimate $J_\pi$. Noting that since $J_\pi = \sum_{s,a} d_\pi(s,a)\phi(s,a)^\top w$, we only need to estimate $e_\pi := \sum_{s,a} d_\pi(s,a)\phi(s,a)$ rather than the full distribution. Under the linear MDP assumption, $e_\pi^\top = b_\pi^\top (I - M_\pi)^{-1}$. Thus, given estimates of the model and reward parameters $\widehat{M}_\pi, \hat{b}_\pi, \hat{w}$, we can evaluate the policy as

$$\widehat{J}_\pi = \hat{b}_\pi^\top (I - \widehat{M}_\pi)^{-1} \hat{w}. \tag{10}$$

### 3.2 OPE error analysis.

Our analysis requires the following assumptions.

**Assumption A1** *(Mixing coefficient)* There exists a constant $\kappa > 0$ such that for any state-action distribution $d$,

$$\left\| (d_\beta - d)^\top \Pi_\beta \right\|_1 \leq \exp(-1/\kappa) \left\| d_\beta - d \right\|_1$$

where $\Pi_\beta$ is the transition matrix from $(s,a)$ to $(s',a')$ under the policy $\beta$.

**Assumption A2** *(Bounded linearly independent features)* Let $\overline{\phi}(s,a)^\top := [\phi(s,a)^\top \ 1]$. We assume that $\max_{s,a} \left\| \overline{\phi}(s,a) \right\|_2 \leq C_\Phi$ for some constant $C_\Phi$. Let $\Phi$ be an $|\mathcal{S}||\mathcal{A}| \times (m+1)$ matrix whose rows are feature vectors $\overline{\phi}(s,a)$. We assume that the columns of $\Phi$ are linearly independent.

**Assumption A3** *(Feature excitation)* For a policy $\pi$ with stationary distribution $d_\pi(s,a)$, define $\Sigma_\pi = \mathbf{E}_{(s,a) \sim d_\pi}[\overline{\phi}(s,a)\overline{\phi}(s,a)^\top]$. We assume that $\lambda_{\min}(\Sigma_\beta) \geq \sigma > 0$ and $\lambda_{\min}(\Sigma_\pi) \geq \sigma_\pi > 0$.

The above assumptions mean that the exploration policy $\beta(a|s)$ mixes fast and is exploratory, in the sense that the stationary distribution spans all dimensions of the feature space. These assumptions allow us to bound the model error. We also require the evaluated policy to span the feature space for somewhat technical reasons, in order to bound the policy evaluation error.

Assume that rewards are linear in the features, $r(s,a) = \phi(s,a)^\top w$. Given a trajectory $\{(s_t, a_t, r_t)\}_{t=1}^{T+1}$, we estimate $M_\pi$, $b_\pi$, and $w$ using regularized least squares:

$$\begin{bmatrix} \widehat{M}_\pi \\ \hat{b}_\pi^\top \end{bmatrix} = \left( \Lambda + \sum_{t=1}^T \overline{\phi}(s_t, a_t)\overline{\phi}(s_t, a_t)^\top \right)^{-1} \sum_{t=1}^T \overline{\phi}(s_t, a_t)\phi(s_{t+1}, \pi)^\top \tag{11}$$

$$\hat{w} = \left( \sum_{t=1}^T \phi(s_t, a_t)\phi(s_t, a_t)^\top \right)^{-1} \sum_{t=1}^T \phi(s_t, a_t)r_t \tag{12}$$

where $\Lambda$ is a regularizer and $\overline{\phi}(s,a) = \begin{bmatrix} \phi(s,a) \\ 1 \end{bmatrix}$. For the purpose of simplifying the analysis, we let $\Lambda = \alpha \sum_{t=1}^T \overline{\phi}(s_t, a_t)\overline{\phi}(s_t, a_t)^\top$; in practice it may be better to use a diagonal matrix. Let $W_\pi = \begin{bmatrix} M_\pi & \mathbf{0} \\ b_\pi^\top & 1 \end{bmatrix}$ and similarly $\widehat{W}_\pi = \begin{bmatrix} \widehat{M}_\pi & \mathbf{0} \\ \hat{b}_\pi^\top & 1 \end{bmatrix}$. The following Lemma (proven in Appendix B) bounds the estimation error under Assumptions A1 and A3 for single-trajectory data:

**Lemma 3.1.** *Let assumptions A1, A2, and A3 hold, and let* $\alpha = C_\Phi^2 \sigma^{-1} \kappa / \sqrt{T}$. *Then with probability at least* $1 - \delta$, *for constants $C$ and $C_w$,*

$$\left\| \widehat{W}_\pi - W_\pi \right\|_2 \leq C C_\Phi^4 \kappa \sigma^{-2} \sqrt{2\ln(2(m+1)/\delta)/T}$$

$$\left\| w - \hat{w} \right\|_2 \leq C_w C_\Phi^2 \kappa \sigma^{-2} \sqrt{2\ln(2m/\delta)/T} \left\| w \right\|_2.$$

The following theorem bounds the policy evaluation error.

**Theorem 3.2** (Policy evaluation error). *Let assumptions A1, A2, and A3 hold and assume that problem (4)-(5) is feasible. Then, for a constant $C_J$, with probability at least $1 - \delta$, the batch policy evaluation error is bounded as*

$$|J_\pi - \widehat{J}_\pi| \leq C_J C_\Phi^4 \kappa \sigma_\pi^{-1/2} \sigma^{-2}(1+\alpha)^2 \sqrt{2\ln(2(m+1)/\delta)/T} \left\| w \right\|_2. \tag{13}$$

The proof is given in Appendix C and relies on expressing evaluation error in terms of the model error, as well as on the contraction properties of the matrix $(1 + \alpha)^{-1} W_\pi$. While we do not provide a lower bound, note that the error scales similarly to the results of Duan and Wang [2020] for discounted MDPs, which nearly match the corresponding lower bound.

**Remark 1**. Theorem 3.2 holds for any feasible solution $\mu$ of (4) and not necessarily just for the maximum-entropy distribution.

**Remark 2**. Our results are shown for the case of discrete states and actions and bounded-norm features. In the continuous case, similar conclusions would follow by arguments on the concentration and boundedness of $\Sigma_\beta$ and $\mathbf{E}_{d_\beta, P}[\overline{\phi}(s, a)\overline{\phi}(s', \pi)^\top]$.

## 4   Related work

The linear MDP assumption along with linear rewards implies that all value functions are linear. Thus we first discuss similarities between our approach and common linear action-value function methods in literature, and then give an broader overview of other related work.

**TD error.** The residual gradient algorithm of Baird [1995] minimizes the mean squared temporal difference (TD) error:

$$L_{TD}(v, J) = \frac{1}{T} \sum_{t=1}^{T} \left( \phi(s_t, a_t)^\top v + J - r_t - \phi(s'_t, \pi)^\top v \right)^2 \tag{14}$$

It is well-known that this objective is a biased and inconsistent estimate of the true Bellman error [Bradtke and Barto, 1996]. Correcting the bias requires double samples (two independent samples of $s'$ for the same $(s, a)$ pair), which may not be available in a single-trajectory dataset. More recent methods rely on fixed point iterations, using the parameters from the previous iteration to construct regression targets. In our setting, fitted Q-iteration (FQI) can be written as

$$v^{(k+1)}, J^{(k+1)} = \arg\min_{v, J} \sum_{t=1}^{T} (\phi(s_t, a_t)^\top v + J - r_t - \phi(s'_t, \pi)^\top v^{(k)})^2 \tag{15}$$

The convergence of FQI is guaranteed only in restricted cases (e.g. Antos et al. [2008]), and no guarantees exist in the undiscounted setting to the best of authors' knowledge.

**PBE error.** Another class of methods minimize the projected Bellman error, which corresponds to only the error representable by the features. The advantage of this approach is that the error due to not having exact dynamics expectations is not correlated with the TD error [Sutton et al., 2009]. Let $D_\beta = \text{diag}(d_\beta)$, and let $Q_\pi$ be the action-value function of $\pi$ as a vector. In matrix form, the projected Bellman equation (PBE) can be written as

$$Q_\pi = G_\beta(r - J_\pi \mathbf{1} + \Pi_\pi Q_\pi), \quad \text{where} \quad G_\beta = \Phi(\Phi^\top D_\beta \Phi)^{-1} \Phi^\top D_\beta \tag{16}$$

where $\Phi$ excludes bias. Methods that attempt to solve (the sample-based version of) the above equation using fixed-point iteration may diverge if the matrix $G_\beta \Pi_\pi$ is not contractive (has spectral radius greater than or equal to 1). The contractiveness condition has been shown to hold in the on-policy setting ($\beta = \pi$) under mild assumptions by Yu and Bertsekas [2009], but need not hold in general. Practical solutions may ensure convergence by regularizing $G_\beta$ as $G_\beta^b = \Phi(\Phi^\top D_\beta \Phi + bI)^{-1} \Phi^\top D_\beta$; however, the required bias $b$ may be large. Another approach, popular in the discounted setting, is to minimize the PBE error using least squares methods like LSTD and LSPE [Yu, 2010a, Lazaric et al., 2010, Dann et al., 2014, Geist and Scherrer, 2014] or using GTD [Liu et al., 2015]. A general complication in the average-reward case is that, unlike with discounted methods which only estimate $Q_\pi$, we also need to solve for $J_\pi$. One possible heuristic is to initially use a guess for $J_\pi$. However, the resulting projected equation may not have a solution [Yu, 2010a, Puterman, 2014], and for OPE, $J_\pi$ is actually the quantity we are after.

**DualDICE** [Nachum and Dai, 2020]. DualDICE and related methods optimize the Lagrangian of the following linear programming (LP) formulation of the Bellman equation:

$$\min_{J, Q} -J \quad \text{s.t.} \quad Q + J\mathbf{1} \leq r + \Pi_\pi Q.$$

For the average reward case, linear function approximation $Q = \Phi v$, and a linear MDP satisfying $\Pi_\pi \Phi = \Phi W_\pi$, the problem Lagrangian is (see also Zhang et al. [2020], Nachum and Dai [2020]):

$$\max_d \min_{J,v} L(d, J, v) = -J + d^\top (\Phi(I - W_\pi)v + J).$$ (17)

We solve an entropy-regularized version of the dual LP using a learned model $\widehat{W_\pi}$. This can be seen as a particular convex instantiation of DualDICE with function approximation - a linear feature model, linear value functions, and exponential-family stationary distributions. See Yang et al. [2020] for different variations of off-policy evaluation via the regularized Lagrangian.

**Dual approaches to batch RL.** Many recent methods for batch policy evaluation and optimization rely on estimating stationary distribution ratios that (approximately) respect the MDP dynamics [Liu et al., 2018, Nachum et al., 2019a,b, Wen et al., 2020]. In particular, Liu et al. [2018] impose a similar constraint to ours on matching feature expectations. However, while we enforce the constraint for a particular feature representation, they minimize the squared error of violating the constraint while maximizing over smooth feature functions in a reproducing kernel Hilbert space. Uehara and Jiang [2020] modify the approach of Liu et al. [2018] to estimate state-action weighs (rather than just state weights) and relax assumptions on the behavior distribution. Feng et al. [2019] minimize a kernel loss for solving the Bellman equation. We note that our approach can also be kernelized, by using kernel ridge regression in place of linear regression for the model. Most of the existing approaches yield consistent estimators, but have no finite-sample guarantees. One exception is the work of Duan and Wang [2020], which provides a minimax lower bound and nearly-matching finite-sample error bound in linear finite-horizon and discounted MDPs, given a dataset of i.i.d. trajectories. Uehara and Jiang [2020] show sample complexity results in a more general setting for discounted ergodic MDPs. Our work provides an analysis for the average-reward linear ergodic setting.

**Maximum-entropy estimation.** The maximum-entropy principle has been well-studied in supervised learning (see e.g. Jaakkola et al. [2000]). There the objective is to maximize the entropy of a distribution subject to feature statistics matching on the available data, and the corresponding dual is maximum-likelihood estimation of an exponential family. In the batch RL setting, we maximize entropy subject to feature expectations matching under the MDP dynamics. For the linear MDP, the resulting distribution is also in the exponential family, and parameterized in a particular way that includes the model. Existing methods for modeling stationary distributions with function approximation tend to use linear functions and require extra constraints to ensure non-negativity and normalization [Rivera Cardoso et al., 2019, Abbasi-Yadkori et al., 2019b]. Exponential families seem like a more elegant solution, and also correspond to well-studied settings such as the linear quadratic regulator. Hazan et al. [2019] proposed learning maximum-entropy stationary distributions for the purpose of exploration. They focused on the tabular MDP case, and required an oracle for solving planning problems with function approximation, since in that case the entropy maximization problem may not be convex. We provide a convex formulation of this problem with function approximation in the linear MDP setting, which can also be used with neural networks (by learning represenations).

## 5 Experiments

We compare our approach to other policy evaluation methods relying on function approximation. Since our focus is not on learning representations, we experiment with a fixed linear basis. We evaluate fitted Q-iteration (FQI) implemented as in (15) and Bellman residual minimization (BRM) implemented as in (14). We also use the average reward of the behavior policy as the simplest baseline. We refer to the closed-form version of our approach in (10) as MODEL, and to the version solving for the stationary distribution as MAXENT. We regularize the covariances of all regression problems using $\alpha I$ with tuned $\alpha$.[3] For MAXENT, we optimize the parameters using full-batch Adam [Kingma and Ba, 2014], and normalize the distributions empirically. For experiments with OpenAI Gym environments [Brockman et al., 2016] (Taxi and Acrobot), we additionally use weighted importance sampling [Mahmood et al., 2014] for both the gradients and the objective. Unless stated otherwise, we generate policies by partially training on-policy using the POLITEX algorithm [Abbasi-Yadkori et al., 2019a], a version of regularized policy iteration with linear Q-functions. We compute the true

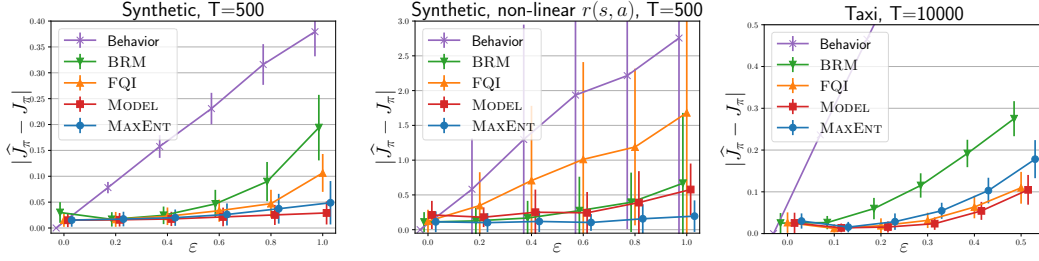

Figure 1: Experiments with behavior policy $\varepsilon$-greedy w.r.t. $\pi$ on synthetic environments and Taxi (mean and standard deviation for 100 target policies $\pi$). Note that the plots are slightly shifted along the horizontal axis to make error bars easier to see.

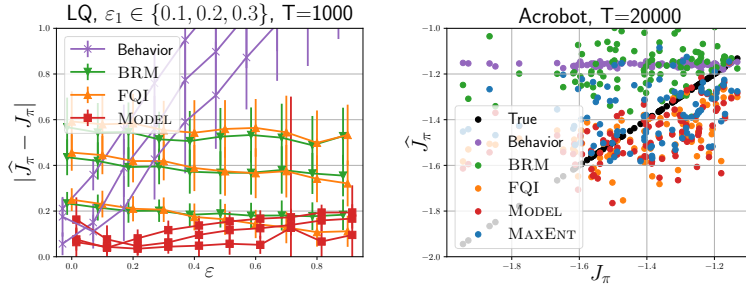

Figure 2: Left: experiments on LQ control, where $\varepsilon$ and $\varepsilon_1$ control the suboptimality of the behavior and target policies, respectively. The more suboptimal policies are difficult to evaluate with BRM and FQI. Right: predicted vs. true value on Acrobot for 100 target policies evaluated using the same behavior policy. The errors are: Behavior 0.24 ($\pm 0.17$), BRM 0.23 ($\pm 0.17$), FQI 0.14 ($\pm 0.10$), MODEL 0.13 ($\pm 0.11$), MAXENT 0.15 ($\pm 0.14$).

policy values $J_\pi$ using Monte-Carlo simulation for Acrobot, and exactly for other environments. Overall, we find that using a feature model is helpful with linear value-function methods.

**Synthetic environments.** We generate synthetic MDPs with 100 states, 10 actions, and transition matrices $P$ generated by sampling entries uniformly at random and normalizing columns to sum to 1. We represent each state with a 10-dimensional vector $\phi_{\mathcal{S}}(s)$ of random Fourier features [Rahimi and Recht, 2008], and let $\phi(s, a) = \phi_{\mathcal{S}}(s) \otimes \phi_{\mathcal{A}}(a)$, where $\phi_{\mathcal{A}}(a)$ is a binary indicator vector for action $a$. We experiment with linear rewards $r(s, a) = -\phi(s, a)^\top w$ with entries of $w$ generated uniformly at random, and with non-linear rewards of the form $r(s, a) = -\exp(2\phi(s, a)^\top w)$. We generate target policies $\pi$ by training on-policy, and set behavior policies $\beta$ to be $\varepsilon$-greedy with respect to $\pi$. We plot the evaluation error $|J_\pi - \widehat{J}_\pi|$ for several values of $\varepsilon$ in Figure 1 (showing mean and standard deviation for 100 random MDPs for each $\varepsilon$). We can see that the model-based approaches are less sensitive to the difference between $\pi$ and $\beta$, and the advantage of inferring the full distribution in the non-linear reward case. Note also that the true underlying dynamics are not low-rank, but our low-rank approximation still results in good estimates.

**Taxi.** The Taxi environment [Dietterich, 2000] is a $5 \times 5$ grid with four pickup/dropoff locations. Taxi actions include going left, right, up, and down, and picking up or dropping off a passenger. There is a reward of -1 for every step, a reward of -10 for illegal pickup/dropoff actions, and a reward of 20 for a successful dropoff. In the infinite-horizon version, a new passenger appears after a successful dropoff. Our state features include indicators for whether the taxi is empty / at pickup / at dropoff and their pairwise products, and xy-coordinates of the taxi, passenger, and dropoff. We set $\pi$ to be 0.05-greedy w.r.t. a hand-coded optimal strategy, and $\beta$ to be $\varepsilon$-greedy w.r.t. $\pi$. A comparison of different policy evaluation methods is given in Figure 1. In this case, all methods are somewhat affected by the suboptimality of the behavior policy, possibly due to fewer successful dropoffs, and FQI and MODEL perform best.

**Linear quadratic regulator.** We evaluate our approach on the linear quadratic (LQ) control system in Dean et al. [2019], where stationary distributions, policies, and transition dynamics are Gaussian. We only evaluate a model-based approach here (as well as FQI and BRM) since the model fully constrains the solution, and solve a constrained optimization problem to ensure positive-definite covariances (see Appendix E for full details). We generate policies by solving the optimal control problem for the true dynamics and noisy costs, where $\varepsilon$ controls the noise for $\beta$ and $\varepsilon_1$ controls the noise for $\pi$. The results are shown in Figure 2 (left) for ten values of $\varepsilon$ and three values of $\varepsilon_1$. While $\varepsilon$ does not seem to affect the OPE performance, the error increases with $\varepsilon_1$ for BRM and FQI.

**Acrobot** [Sutton, 1996] is a simple episodic discrete-action physical control task. The system includes two links and two joints, one of which is actuated. The goal is to swing the lower link up to a given height. We set the reward at each time step to the negative distance between the joint to its target height, and to 100 when the lower link reaches its target height. Each episode ends after 500 steps, or after the target height is reached, after which we reset. The observations are link positions and velocities; we featurize them using the multivariate Fourier basis of order 3 as described in Konidaris et al. [2011]. For this task, we partially train 101 policies, set $\beta$ to the first policy, and evaluate the remaining policies. The results are shown in Figure 2 (right). In this case, BRM predictions seem more correlated to the behavior policy than to the target. The other methods are better correlated with the target policy, but have somewhat high error. Possible reasons for this are the episodic nature of the environment, and the true underlying dynamics being only locally linear.

## 6   Conclusion and future work

We have presented a new approach to batch policy evaluation in average-reward MDPs. For linear MDPs, we have provided a finite-sample bound on the OPE error, which extends the previous results for discounted and episodic settings. In a more general setting with non-linear rewards and approximately linear feature dynamics, we have proposed a maximum-entropy approach to finding stationary distributions with function approximation. One weakness of our analysis is that it requires both the behavior and target policy to induce an exploratory distribution in order to produce reliable model estimates. We hope to remove this assumption in future work. Given that the linear MDP assumption is fairly restrictive, an important direction for future work is extending the framework beyond linear functions. Another direction to explore is applying this framework to policy optimization. Finally, note that the maximum-entropy objective corresponds to minimizing the KL-divergence between the target distribution and the uniform distribution, and we can easily minimize the KL divergence to other distributions instead. While the maximum entropy objective is justified in some cases (see Appendix D), our formulation allows us to incorporate other prior knowledge and constraints when available, and this is another direction for future work.

### Broader impact

In general, when learning from a batch of data produced by a fixed behavior policy, we may inherit the biases of that policy, and our models may not generalize beyond the support of the data distribution. In our paper, we circumvent this issue by assuming that the information sufficient for evaluating and optimizing policies is contained in some known features, and that the behavior policy is exploratory enough in the sense that it spans those features. These assumptions may not always hold when applying the method in practice.

### Funding disclosure

This work was funded through the authors' employment by Alphabet Inc. Dale Schuurmans is also employed by the University of Alberta. No authors have any competing interests.

## Footnotes

*DeepMind

†Google

[3]Starting with $\alpha = 1$, we keep doubling $\alpha$ for FQI as long as it diverges, and for MAXENT as long as $|\lambda_{\max}(\widehat{W_\pi})| > 1$.

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
