[Supplementary Material]

# A  Off-policy evaluation dual objective

We formulate the estimation of the stationary state distribution $\mu_\pi(s)$ given a policy $\pi(a|s)$ as a maximum-entropy problem subject to matching feature expectations under the linearity assumption:

$$\min_\mu \ \sum_s \mu(s) \ln \mu(s) \tag{18}$$

$$\text{s.t.} \ \sum_{s,a} \mu(s)\pi(a|s)\phi(s,a)^\top(I - M_\pi) = b_\pi^\top \tag{19}$$

$$\sum_s \mu(s) = 1 \tag{20}$$

Let $\phi(s,\pi) = \sum_a \pi(a|s)\phi(s,a)$ be the expected state features under the target policy. For a fixed / given $M_\pi$, the Lagrangian of the above objective is:

$$L(\mu, \theta, \lambda) = \sum_s \mu(s) \ln \mu(s) + b_\pi^\top \theta - \sum_s \mu(s)\phi(s,\pi)^\top(I - M_\pi)\theta - \lambda(\sum_s \mu(s) - 1)$$

Setting the gradient of $L(\mu, \theta)$ w.r.t. $\mu(s)$ to zero, we get

$$0 = \ln \mu(s) + 1 - \phi(s,\pi)^\top(I - M_\pi)\theta - \lambda$$

$$\mu(s) = \exp(\phi(s,\pi)^\top(I - M_\pi)\theta + \lambda - 1)$$

Because $\sum_s \mu(s) = 1$, we get

$$1 - \lambda = \ln \sum_s \exp(\phi(s,\pi)^\top(I - M_\pi)\theta) := F(\theta|M_\pi),$$

where $F(\theta|M_\pi)$ is the log-normalizer. By plugging this expression for $\mu$ into the Lagrangian, we get the following dual maximization objective in $\theta$:

$$D(\theta) := \sum_s \mu(s)(\phi(s,\pi)^\top(I - M_\pi)\theta - F(\theta|M_\pi)) + b_\pi^\top \theta - \sum_s \mu(s)\phi(s,\pi)^\top(I - M_\pi)\theta$$

$$= b_\pi^\top \theta - F(\theta|M_\pi).$$

# B  Model error

## B.1  Preliminaries

Our error analysis relies on similar techniques as the finite-sample analysis in Abbasi-Yadkori et al. [2019a]. We first state some useful results.

**Lemma B.1** (Lemma A.1 in [Abbasi-Yadkori et al., 2019a])**.** *Let Assumption A1 hold, and let $\{(s_t, a_t)\}_{t=1}^T$ be the state-action sequence obtained when following the behavior policy $\beta$ from an initial distribution $d_0$. For $t \in [T]$, let $X_t$ be a binary indicator vector with a non-zero at the linear index of the state-action pair $(s_t, a_t)$. Define for $i \in [T]$,*

$$B_i = \mathbf{E}\left[\sum_{t=1}^T X_t | X_1, ..., X_i\right], \quad \text{and} \ \ B_0 = \mathbf{E}\left[\sum_{t=1}^T X_t\right].$$

*Then, $(B_i)_{i=0}^T$ is a vector-valued martingale: $\mathbf{E}[B_i - B_{i-1}|B_0, \dots, B_{i-1}] = 0$ for $i = 1, \dots, T$, and $\|B_i - B_{i-1}\|_1 \le 4\kappa$ holds for $i \in [T]$.*

The constructed martingale is known as the Doob martingale underlying the sum $\sum_{t=1}^T X_t$. Let $\Pi_\beta$ be the transition matrix for state-action pairs when following $\beta$. Then, for $t = 0, \dots, m-1$, $\mathbf{E}[X_{t+1}|X_t] = \Pi_\beta^\top X_t$ and by the Markov property, for any $i \in [T]$,

$$B_i = \sum_{t=1}^i X_t + \sum_{t=i+1}^T \mathbf{E}[X_t|X_i] = \sum_{t=1}^i X_t + \sum_{t=1}^{T-i} (\Pi_\beta^t)^\top X_i \quad \text{and} \quad B_0 = \sum_{t=1}^T (\Pi_\beta^t)^\top X_0.$$

It will be useful to define another Doob martingale as follows:

$$Y_i = \sum_{t=2}^{i} X_{t-1}X_t^\top + \sum_{t=i+1}^{T} \mathbf{E}[X_{t-1}X_t^\top | X_i] = \sum_{t=2}^{i} X_{t-1}X_t^\top + \sum_{t=i+1}^{T} \mathrm{diag}(X_i^\top \Pi_\beta^{t-1})\Pi_\beta \quad (21)$$

$$Y_0 = \sum_{t=1}^{T} \mathbf{E}[X_{t-1}X_t^\top] = \sum_{t=1}^{T} \mathrm{diag}(d_0^\top \Pi_\beta^{t-1})\Pi_\beta \quad (22)$$

where $d_0$ is the initial state-action distribution. The difference sequence can again be bounded as $\|Y_i - Y_{i-1}\|_{1,1} \leq 4\kappa$ under the mixing assumption (see Appendix D.2.2 of Abbasi-Yadkori et al. [2019a] for more details).

Let $(\mathcal{F}_k)_k$ be a filtration and define $\mathbf{E}_k[\cdot] := \mathbf{E}[\cdot|\mathcal{F}_k]$.

**Theorem B.2** (Matrix Azuma [Tropp, 2012]). *Consider a finite $(\mathcal{F})_k$-adapted sequence $\{X_k\}$ of Hermitian matrices of dimension $m$, and a fixed sequence $\{A_k\}$ of Hermitian matrices that satisfy $\mathbf{E}_{k-1}X_k = 0$ and $X_k^2 \preceq A_k^2$ almost surely. Let $v = \|\sum_k A_k^2\|$. Then for all $t \geq 0$,*

$$P\left(\lambda_{\max}\left(\sum_k X_k\right) \geq t\right) \leq m \cdot \exp(-t^2/8v).$$

Equivalently, with probability at least $1-\delta$, $\|\sum_k X_k\| \leq 2\sqrt{2v \ln(m/\delta)}$. A version of the inequality for non-Hermitian matrices of dimension $m_1 \times m_2$ can be obtained by applying the theorem to a Hermitian dilation of $X$, $\mathcal{D}(X) = \begin{bmatrix} 0 & X \\ X^* & 0 \end{bmatrix}$, which satisfies $\lambda_{\max}(\mathcal{D}(X)) = \|X\|$ and $\mathcal{D}(X)^2 = \begin{bmatrix} XX^* & 0 \\ 0 & X^*X \end{bmatrix}$. In this case, we have that $v = \max(\|\sum_k X_k X_k^*\|, \|\sum_k X_k^* X_k\|)$.

Let $\Phi$ be a $|\mathcal{S}||\mathcal{A}| \times (m+1)$ matrix whose rows correspond to bias-augmented feature vectors $\overline{\phi}(s,a)$ for each state-action pair $(s,a)$. Let $\phi_i$ be the feature vector corresponding to the $i^{th}$ row of $\Phi$, and let $C_\Phi = \max_i \|\phi_i\|_2$. For any matrix $A$, we have

$$\|\Phi^\top A \Phi\|_2 = \left\|\sum_{ij} A_{ij}\phi_i\phi_j^\top\right\|_2 \leq \sum_{i,j} |A_{ij}| \|\phi_i\phi_j^\top\|_2 \leq C_\Phi^2 \sum_{i,j} |A_{ij}| = C_\Phi^2 \|A\|_{1,1}. \quad (23)$$

## B.2 Proof of Lemma 3.1

*Proof.* Let $\Phi$ be a $|\mathcal{S}||\mathcal{A}| \times (m+1)$ matrix whose rows correspond to bias-augmented feature vectors $\overline{\phi}(s,a)$. Let $D_\beta = \mathrm{diag}(d_\beta)$. Let $\tilde{d}_\beta$ be the empirical data distribution, and $\tilde{D}_\beta = \mathrm{diag}(\tilde{d}_\beta)$. The true and estimated (concatenated) model parameters can be written as

$$\widehat{W}_\pi = (\Lambda + \Phi^\top \tilde{D}_\beta \Phi)^{-1} \Phi^\top \tilde{D}_\beta \tilde{\Pi}_\pi \Phi$$

$$W_\pi = (\Phi^\top D_\beta \Phi)^{-1} \Phi^\top D_\beta \Pi_\pi \Phi$$

where $\Pi_\pi$ is the true state-action transition kernel under $\pi$, and $\tilde{\Pi}_\pi$ corresponds to empirical next-state dynamics $\tilde{P}$. For the true model satisfying $\Pi_\pi \Phi = \Phi M_\pi$, we have taken expectations over $d_\beta$ and taken advantage of Assumption A3.

Let $\Lambda = \alpha \Phi^T \tilde{D}_\beta \Phi$; in this case

$$\widehat{W}_\pi = \frac{1}{1+\alpha} (\Phi^\top \tilde{D}_\beta \Phi)^{-1} \Phi^\top \tilde{D}_\beta \tilde{\Pi}_\pi \Phi$$

We first bound the error for $(1+\alpha)\widehat{M}_\pi$. The model error can be upper-bounded as:

$$\left\|(1+\alpha)\widehat{W}_\pi - W_\pi\right\|_2 \leq \left\|(\Phi^\top D_\beta \Phi)^{-1}\Phi^\top(\tilde{D}_\beta\tilde{\Pi}_\pi - D_\beta\Pi_\pi)\Phi\right\|_2$$

$$+ \left\|((\Phi^\top \tilde{D}_\beta \Phi)^{-1} - (\Phi^\top D_\beta \Phi)^{-1})\Phi^\top \tilde{D}_\beta \Pi_\pi \Phi\right\|_2$$

$$\leq \sigma^{-1}\left\|\Phi^\top(\tilde{D}_\beta - D_\beta)\Pi_\pi\Phi\right\|_2 + \left\|(\Phi^\top \tilde{D}_\beta \Phi)^{-1} - (\Phi^\top D_\beta \Phi)^{-1}\right\|_2 \left\|\Phi^\top \tilde{D}_\beta \Pi_\pi \Phi\right\|_2$$

$$\leq \sigma^{-1} \left\| \Phi^\top (\tilde{D}_\beta - D_\beta)\Pi_\pi \Phi \right\|_2 + C_\Phi^2 \left\| (\Phi^\top \tilde{D}_\beta \Phi)^{-1} - (\Phi^\top D_\beta \Phi)^{-1} \right\|_2$$

where the second inequality follows from Assumption A3, and the last inequality follows from (23). We proceed to bound the two terms

$$E_1 = \sigma^{-1} \left\| \Phi^\top (\tilde{D}_\beta \tilde{\Pi}_\pi - D_\beta \Pi_\pi)\Phi \right\|_2$$

$$E_2 = \left\| (\Phi^\top \tilde{D}_\beta \Phi)^{-1} - (\Phi^\top D_\beta \Phi)^{-1} \right\|_2$$

**Bounding $E_1$.** Let $(Y_i)_i$ be the Doob martingale defined in (21)-(22), and let $\tilde{\Pi}_\beta$ be the empirical state-action transition matrix under the policy $\beta$. Note that $\tilde{D}_\beta \tilde{\Pi}_\beta = Y_T/T$. Furthermore, let $K^\pi$ be a $|\mathcal{S}||\mathcal{A}| \times |\mathcal{S}||\mathcal{A}|$ matrix defined as

$$K^\pi_{(s,a),(s',a')} = \begin{cases} \pi(a'|s) & \text{if } s' = s \\ 0 & \text{otherwise} \end{cases}$$

Notice that $\tilde{D}_\beta \tilde{\Pi}_\beta K^\pi = \tilde{D}_\beta \tilde{\Pi}_\pi$ and $D_\beta \Pi_\beta K^\pi = D_\beta \Pi_\pi$. We can upper-bound $E_1$ as:

$$\sigma^{-1} \left\| \Phi^\top (\tilde{D}_\beta \tilde{\Pi}_\beta - D_\beta \Pi_\beta)K^\pi \Phi \right\|_2 = \frac{1}{\sigma T} \left\| \Phi^\top (Y_T - Y_0)K^\pi \Phi \right\|_2 + \frac{1}{\sigma T} \left\| \Phi^\top (Y_0 - TD_\beta \Pi_\beta)K^\pi \Phi \right\|_2$$

Note that $\Phi^\top Y_i K^\pi \Phi$ is a matrix-valued martingale, whose difference sequence is bounded by

$$\left\| (\Phi^\top (Y_i - Y_{i-1})K^\pi \Phi)^2 \right\|_2 \leq C_\Phi^4 \left\| (Y_i - Y_{i-1})K^\pi \right\|_{1,1}^2 \leq 16 C_\Phi^4 \kappa^2$$

where we have used (23) and the fact that rows of $K^\pi$ sum to 1. Applying the matrix-Azuma theorem B.2, we have that with probability at least $\delta$,

$$\frac{1}{\sigma T} \left\| \Phi^\top (Y_T - Y_0)K^\pi \Phi \right\|_2 \leq 8 C_\Phi^2 \sigma^{-1} \kappa \sqrt{2\ln(2(m+1)/\delta)/T} .$$

Using the mixing Assumption A1, and letting $d_0$ be the initial state-action distribution,

$$\frac{1}{\sigma T} \left\| \Phi^\top (Y_0 - TD_\beta \Pi_\beta)K^\pi \Phi \right\|_2 \leq \frac{1}{\sigma T} \sum_{t=1}^T \left\| \Phi^\top \mathrm{diag}(d_0^\top \Pi_\beta^t - d_\beta^\top)\Pi_\beta K^\pi \Phi \right\|_2$$

$$\leq \frac{C_\Phi^2}{\sigma T} \sum_{t=1}^T \left\| \mathrm{diag}(d_0^\top \Pi_\beta^t - d_\beta^\top)\Pi_\pi \right\|_{1,1}$$

$$\leq \frac{C_\Phi^2}{\sigma T} \sum_{t=1}^T \exp(-t/\kappa) \left\| d_0 - d_\beta \right\|_1 \leq \frac{2 C_\Phi^2 \kappa}{\sigma T}$$

Thus we get that with probability at least $1 - \delta$,

$$E_1 \leq 8 C_\Phi^2 \sigma^{-1} \kappa \left( \sqrt{2\ln(2(m+1)/\delta)/T} + 1/T \right)$$

**Bounding $E_2$.** To bound $E_2$, we first rely on the Woodbury identity to write

$$(\Phi^\top \tilde{D}_\beta \Phi)^{-1} - (\Phi^\top D_\beta \Phi)^{-1}$$

$$= (\Phi^\top D_\beta \Phi + \Phi^\top (D_\beta - \tilde{D}_\beta)\Phi)^{-1} - (\Phi^\top D_\beta \Phi)^{-1}$$

$$= (\Phi^\top D_\beta \Phi)^{-1} \big( (\Phi^\top D_\beta \Phi)^{-1} + (\Phi^\top (\tilde{D}_\beta - D_\beta)\Phi)^{-1} \big)^{-1} (\Phi^\top D_\beta \Phi)^{-1}$$

$$E_2 \leq \sigma^{-2} \left\| \big( (\Phi^\top D_\beta \Phi)^{-1} + (\Phi^\top (\tilde{D}_\beta - D_\beta)\Phi)^{-1} \big)^{-1} \right\|_2$$

$$\leq \sigma^{-2} \left\| \Phi^\top (\tilde{D}_\beta - D_\beta)\Phi \right\|_2$$

$$\leq 8 \sigma^{-2} C_\Phi^2 \kappa \left( \sqrt{2\ln(2(m+1)/\delta)/T} + 1/T \right)$$

where the second line follows because $(\Phi^\top D_\beta \Phi)^{-1} \succ 0$, and the last line follows by similar concentration arguments as those for $E_1$ for the matrix-valued martingale $\Phi^\top \mathrm{diag}(B_i)\Phi$, with probability at least $1 - \delta$.

**Bounding** $\left\| \widehat{W}_\pi - W_\pi \right\|_2$. Putting previous terms together, with probability at least $1 - \delta$, for an absolute constant $C$,

$$\left\| (1 + \alpha)\widehat{W}_\pi - W_\pi \right\|_2 \leq C C_\Phi^4 \kappa \sigma^{-2} \sqrt{2 \ln(2(m+1)/\delta)/T} \tag{24}$$

Furthermore, we have:

$$\begin{aligned}
\left\| \widehat{W}_\pi - W_\pi \right\|_2 &\leq \left\| (1+\alpha)\widehat{W}_\pi - W_\pi \right\|_2 + \alpha \left\| \widehat{W} \right\|_2 \\
&\leq C C_\Phi^4 \kappa \sigma^{-2} \sqrt{2 \ln(2(m+1)/\delta)/T} + \alpha \sigma^{-1} C_\Phi^2
\end{aligned}$$

Setting $\alpha = C_\Phi^2 \sigma^{-1} \kappa / \sqrt{T}$ gives the final result.

**Bounding** $\|w - \hat{w}\|$. For linear rewards $r(s, a) = \phi(s, a)^\top w$, we estimate the parameters $w$ using linear regression. Abusing notation, assume that the feature matrix $\Phi$ does not include bias for the purpose of this section. The true and estimated parameters $w$ and $\hat{w}$ satisfy

$$w = (\Phi^\top D_\beta \Phi)^{-1} \Phi^\top D_\beta r \tag{25}$$

$$\hat{w} = (\Phi^\top \tilde{D}_\beta \Phi)^{-1} \Phi^\top \tilde{D}_\beta r \tag{26}$$

where $r = \Phi w$ is the length-$|\mathcal{S}||\mathcal{A}|$ vector of rewards. We have that

$$\|w - \hat{w}\| = \left\| (\Phi^\top D_\beta \Phi)^{-1} \Phi^\top (D_\beta - \tilde{D}_\beta) \Phi w \right\| + \left\| ((\Phi^\top D_\beta \Phi)^{-1} - (\Phi^\top \tilde{D}_\beta \Phi)^{-1}) \Phi^\top \tilde{D}_\beta \Phi w \right\| \tag{27}$$

Using the bounds from the previous section, we get that for a constant $C_w$, with probability at least $1 - \delta$,

$$\|w - \hat{w}\| \leq C_w C_\Phi^2 \sigma^{-2} \kappa (\sqrt{2 \ln(2m/\delta)/T}) \|w\| \tag{28}$$

$\square$

## C   Proof of Theorem 3.2 (policy evaluation error)

*Proof.* Assuming that the optimization problem is feasible, the following holds for the resulting distribution $\hat{d}_\pi(s, a) = \hat{\mu}_\pi(s)\pi(a|s)$:

$$\sum_{s,a} \hat{d}_\pi(s, a) \overline{\phi}(s, a)^\top = \sum_{s,a} \hat{d}_\pi(s, a) \overline{\phi}(s, a)^\top \widehat{W}_\pi \, .$$

Assume that the reward is linear in the features, $r(s, a) = w^\top \phi(s, a)$, and let $\hat{w}$ be the corresponding parameter estimate. Let $\underline{w} = \left[ \begin{smallmatrix} w \\ 0 \end{smallmatrix} \right]$ and let $\underline{\hat{w}} = \left[ \begin{smallmatrix} \hat{w} \\ 0 \end{smallmatrix} \right]$.

The policy evaluation error is:

$$\begin{aligned}
J_\pi - \widehat{J}_\pi &= \sum_{s,a} (d_\pi(s, a) \overline{\phi}(s, a)^\top \underline{w} - \hat{d}_\pi(s, a) \overline{\phi}(s, a)^\top \underline{\hat{w}}) \\
&= \sum_{s,a} (d_\pi(s, a) - \hat{d}_\pi(s, a)) \overline{\phi}(s, a)^\top \underline{w} + \sum_{s,a} \hat{d}_\pi(s, a) \phi(s, a)^\top (\underline{w} - \underline{\hat{w}}) \, .
\end{aligned}$$

The norm of the second term is bounded by $C_\Phi \|w - \hat{w}\|_2$. We proceed to bound the first term.

Let $W_\pi^\alpha = \frac{1}{1+\alpha} W_\pi$, and define $e^\top := \sum_{s,a} d(s, a) \overline{\phi}(s, a)^\top$ and $\hat{e}^\top := \sum_{s,a} \hat{d}(s, a) \overline{\phi}(s, a)^\top$. The first term can be written as:

$$(e^\top - \hat{e}^\top)\underline{w} = e^\top (W_\pi^\alpha + \alpha W_\pi^\alpha) w - \hat{e}^\top \widehat{W}_\pi)\underline{w}$$

$$= (e - \hat{e})^\top W_\pi^\alpha \underline{w} + \hat{e}^\top (W_\pi^\alpha - \widehat{W}_\pi)\underline{w} + \alpha e^\top W_\pi^\alpha \underline{w}$$
$$= (e - \hat{e})^\top (W_\pi^\alpha)^2 \underline{w}$$
$$+ \hat{e}^\top (W_\pi^\alpha - \widehat{M}_\pi)(I + M_\pi^\alpha)\underline{w}$$
$$+ \alpha e^\top W_\pi^\alpha (I + (W_\pi^\alpha)^2)\underline{w}$$
$$= \lim_{K \to \infty} (e - \hat{e})^\top (W_\pi^\alpha)^K \underline{w} + \left( \hat{e}^\top (W_\pi^\alpha - \widehat{W}_\pi) + \alpha e^\top W_\pi^\alpha \right) \left( \sum_{i=0}^{K} (W_\pi^\alpha)^i \right) \underline{w}$$

In order to evaluate the infinite sum, we first show that $W_\pi$ is non-expansive in a $\Sigma_\pi$-weighted norm (and hence $W_\pi^\alpha$ is contractive):

$$\Sigma_\pi := \mathbf{E}_{(s,a) \sim d_\pi}[\overline{\phi}(s,a)\overline{\phi}(s,a)^\top]$$
$$= \mathbf{E}_{(s,a) \sim d_\pi}[\mathbf{E}_{(s',a') \sim \Pi_\pi(\cdot|s,a)}[\overline{\phi}(s',a')\overline{\phi}(s',a')^\top]]$$
$$= \mathbf{E}_{(s,a) \sim d_\pi}[W_\pi^\top \overline{\phi}(s,a)\overline{\phi}(s,a)^\top W_\pi] + V$$
$$= W_\pi^\top \Sigma_\pi W_\pi + V \qquad (29)$$

where $V \succeq 0$. Multiplying each side of (29) by $\Sigma_\pi^{-1/2}$ from the left- and right-hand side, we get that

$$I = \Sigma_\pi^{-1/2} W_\pi^\top \Sigma_\pi^{1/2} \Sigma_\pi^{1/2} W_\pi \Sigma_\pi^{-1/2} + \Sigma_\pi^{-1/2} V \Sigma_\pi^{-1/2}$$
$$1 \geq \left\| \Sigma_\pi^{1/2} W_\pi \Sigma_\pi^{-1/2} \right\|_2^2$$

Thus we have that $\left\| \Sigma_\pi^{1/2} W_\pi^\alpha \Sigma_\pi^{-1/2} \right\|_2 \leq (1 + \alpha)^{-1}$, and we can compute the infinite sum as:

$$(W_\pi^\alpha)^i = \Sigma_\pi^{-1/2} \left( \Sigma_\pi^{1/2} W_\pi^\alpha \Sigma_\pi^{-1/2} \right)^i \Sigma_\pi^{1/2}$$
$$\left\| \sum_{i=0}^{\infty} (W_\pi^\alpha)^i \right\| \leq \sum_{i=0}^{\infty} \left\| \Sigma_\pi^{-1/2} \right\| \left\| \Sigma_\pi^{1/2} W_\pi^\alpha \Sigma^{-1/2} \right\|^i \left\| \Sigma_\pi^{1/2} \right\| \leq C_\Phi \sigma_\pi^{-1/2}(1 + \alpha)$$

The error can now be written as

$$|J_\pi - \widehat{J}_\pi| = \left( \|\hat{e}\|_2 \left\| W_\pi^\alpha - \widehat{W}_\pi \right\|_2 + \alpha \left\| e^\top W_\pi^\alpha \right\|_2 + \|u\|_2 \right) C_\Phi \sigma_\pi^{-1/2}(1 + \alpha) \|w\|_2 + C_\Phi \|w - \hat{w}\|_2$$

Note that $\|e\|_2 \leq C_\Phi$ and $\|\hat{e}\|_2 \leq C_\phi$. Set $\alpha = C_\Phi^2 \sigma^{-1} \kappa / \sqrt{T}$ as in the previous section. From (24), we have that

$$\left\| W_\pi^\alpha - \widehat{W}_\pi \right\|_2 \leq (1 + \alpha) C C_\Phi^4 \kappa \sigma^{-2} \sqrt{2 \ln(2(m+1)/\delta)/T} \qquad (30)$$

Plugging in the model errors and $\alpha$ and combining terms we get the final result in the theorem.

$\square$

# D  Stationary distribution with large entropy

In this paper, we try to find the distribution over states that maximizes the entropy under some linear constraints, and use it as a proxy for the stationary distribution. In this section, we provide some theoretical evidence that at least when the probability transition over the states is sufficiently random, the stationary distribution tends to have large entropy.

For simplicity, we focus on finite-state Markov chains instead of MDPs. Consider a Markov chain with state space $\mathcal{S}$ and probability transition matrix $P$. Let $S := |\mathcal{S}|$. Then the stationary distribution $d$ satisfies $d^\top = d^\top P$. In this section, we assume that each row of $P$ is sampled uniformly at random from the simplex over $\mathcal{S}$, i.e., $\Delta_\mathcal{S}$, independently of other rows. We prove the following result, which shows that as $S$ increases, the stationary distribution $d$ converges to a uniform distribution over the states at a rate $\mathcal{O}(1/\sqrt{S})$.

**Theorem D.1.** *Let $P$ be the probability transition matrix of a Markov chain with finite state space $\mathcal{S}$, and assume that rows of $P$ are sampled independently and uniformly at random from $\Delta_{\mathcal{S}}$. Then, with probability at least $1 - \delta$, the stationary distribution $d$ of the Markov chain satisfies*

$$\left\| \frac{d}{\|d\|_2} - \frac{1}{\sqrt{S}} \mathbf{1} \right\|_2 \leq \frac{2\sqrt{10}}{\delta\sqrt{S}},$$

*where $\mathbf{1}$ denotes an all-one vector.*

*Proof.* We denote the uniform distribution over the simplex in $\mathbb{R}^S$ by $\mathcal{U}$. The distribution $\mathcal{U}$ is a special case of Dirichlet distribution [Hazewinkel, 2001]. In this proof, we make use of the following properties of $\mathcal{U}$.

**Lemma D.2.** *[Hazewinkel, 2001] Let $x \sim \mathcal{U}$ and $x_i$ be the $i$-th coordinate of $x$. Then we have*

$$\mathbf{E}[x_i] = \frac{1}{S}, \quad \mathbf{E}[x_i^2] = \frac{2}{S(S+1)}, \quad \mathbf{E}[x_i^4] = \frac{24}{S(S+1)(S+2)(S+3)}$$

$$\mathbf{E}[x_i x_j] = \frac{1}{S(S+1)}, \quad \mathbf{E}[x_i^2 x_j^2] = \frac{4}{S(S+1)(S+2)(S+3)}, \quad \forall i \neq j.$$

This lemma gives us the following direct corollary.

**Corollary D.3.** *Suppose that $x$ and $y$ are two independent samples from $\mathcal{U}$. Then we have*

$$\mathbf{E}[\|x\|_2^2] = \frac{2}{S+1} \tag{31}$$

$$\mathbf{E}[x^\top y] = \frac{1}{S} \tag{32}$$

$$\mathbf{E}[\|x\|_2^4] = \frac{4(S+5)}{(S+1)(S+2)(S+3)} \tag{33}$$

$$\mathbf{E}[(x^\top y)^2] = \frac{S+3}{S(S+1)^2} \tag{34}$$

*Proof.*

$$\mathbf{E}[\|x\|_2^2] = \mathbf{E}\left[ \sum_{i=1}^{S} x_i^2 \right] = \frac{2}{S+1}.$$

$$\mathbf{E}[x^\top y] = \mathbf{E}\left[ \sum_{i=1}^{S} x_i y_i \right] = \sum_{i=1}^{S} \mathbf{E}[x_i]\mathbf{E}[y_i] = \frac{1}{S}.$$

$$\mathbf{E}[\|x\|_2^4] = \mathbf{E}\left[ (\sum_{i=1}^{S} x_i^2)^2 \right] = \sum_{i=1}^{S} \mathbf{E}[x_i^4] + \sum_{i \neq j} \mathbf{E}[x_i^2 x_j^2] = \frac{4(S+5)}{(S+1)(S+2)(S+3)}.$$

$$\mathbf{E}[(x^\top y)^2] = \mathbf{E}\left[ (\sum_{i=1}^{S} x_i y_i)^2 \right] = \sum_{i=1}^{S} \mathbf{E}[x_i^2 y_i^2] + \sum_{i \neq j} \mathbf{E}[x_i x_j y_i y_j] = \frac{S+3}{S(S+1)^2}.$$

$\square$

Now we turn to the proof of Theorem D.1. In the following, we define $\widehat{\Sigma} := PP^\top$, $\Sigma := \mathbf{E}[\widehat{\Sigma}]$, and let $p_i$ be the $i$-th column of $P^\top$. For a PSD matrix $M$, we define $\lambda_i(M)$ as its $i$-th largest eigenvalue. Since $P$ is a probability transition matrix, we know that $\lambda_1(\widehat{\Sigma}) = 1$, and the corresponding top eigenvector is $\frac{d}{\|d\|_2}$. We then analyze $\Sigma$. Since $\Sigma_{i,j} = \mathbf{E}[p_i^\top p_j]$, according to Corollary D.3, we know that $\Sigma_{i,i} = \frac{2}{S+1}, \forall i$ and $\Sigma_{i,j} = \frac{1}{S}, \forall i \neq j$. Thus

$$\Sigma = \frac{S-1}{S(S+1)} I + \frac{1}{S} \mathbf{1}\mathbf{1}^\top.$$

Then, we know that $\lambda_1(\Sigma) = 1 + \frac{S-1}{S(S+1)}$, $\lambda_i(\Sigma) = \frac{S-1}{S(S+1)}$, $\forall i \geq 2$. Then, the gap between the top eigenvalue of $\Sigma$ and the second largest eigenvalue of $\Sigma$ is

$$\lambda_1(\Sigma) - \lambda_2(\Sigma) = 1. \tag{35}$$

The top eigenvector of $\Sigma$ is $\frac{1}{\sqrt{S}}\mathbf{1}$. Next, we proceed to bound the difference between $\Sigma$ and $\widehat{\Sigma}$. In particular, we bound $\mathbf{E}[\|\widehat{\Sigma} - \Sigma\|_F^2]$. We have

$$
\begin{aligned}
\mathbf{E}[\|\widehat{\Sigma} - \Sigma\|_F^2] &= \sum_{i=1}^{S}\left(\mathbf{E}[\widehat{\Sigma}_{i,i}^2] - \mathbf{E}[\widehat{\Sigma}_{i,i}]^2\right) + \sum_{i \neq j}\left(\mathbf{E}[\widehat{\Sigma}_{i,j}^2] - \mathbf{E}[\widehat{\Sigma}_{i,j}]^2\right) \\
&= \sum_{i=1}^{S}\left(\mathbf{E}[\|p_i\|_2^4] - \mathbf{E}[\|p_i\|_2^2]^2\right) + \sum_{i \neq j}\left(\mathbf{E}[(p_i^\top p_j)^2] - \mathbf{E}[p_i^\top p_j]^2\right) \\
&= \frac{4S(S-1)}{(S+1)^2(S+2)(S+3)} + \frac{(S-1)^2}{S(S+1)^2} \tag{36} \\
&\leq \frac{5}{S}, \tag{37}
\end{aligned}
$$

where in (36) we use Corollary D.3. Thus, we have

$$\mathbf{E}[\|\widehat{\Sigma} - \Sigma\|_F] \leq \sqrt{\mathbf{E}[\|\widehat{\Sigma} - \Sigma\|_F^2]} \leq \sqrt{\frac{5}{S}}. \tag{38}$$

According to Markov's inequality, with probability at least $1 - \delta$,

$$\|\widehat{\Sigma} - \Sigma\|_F \leq \frac{\sqrt{5}}{\delta\sqrt{S}}. \tag{39}$$

We then apply Davis-Kahan Theorem [Davis and Kahan, 1970] (see also Theorem 2 in Yu et al. [2015]) and obtain

$$\sqrt{1 - \langle \frac{d}{\|d\|_2}, \frac{1}{\sqrt{S}}\mathbf{1}\rangle^2} \leq \frac{2\|\widehat{\Sigma} - \Sigma\|_F}{\lambda_1(\Sigma) - \lambda_2(\Sigma)} = 2\|\widehat{\Sigma} - \Sigma\|_F,$$

where for the equality we use (35). This implies

$$
\begin{aligned}
\left\|\frac{d}{\|d\|_2} - \frac{1}{\sqrt{S}}\mathbf{1}\right\|_2 &= \sqrt{2 - 2\langle\frac{d}{\|d\|_2}, \frac{1}{\sqrt{S}}\mathbf{1}\rangle} \\
&\leq \sqrt{2}\sqrt{1 - \langle\frac{d}{\|d\|_2}, \frac{1}{\sqrt{S}}\mathbf{1}\rangle^2} \\
&\leq 2\sqrt{2}\|\widehat{\Sigma} - \Sigma\|_F. \tag{40}
\end{aligned}
$$

Then we can complete the proof by combining (39) and (40). $\qquad\square$

## E  Experiment details for linear quadratic control

In a linear-quadratic (LQ) control problem, the dynamics are linear-Gaussian in states $x$:

$$x_{t+1} = Ax_t + Ba_t + w_t, \quad w_t \sim \mathcal{N}(0, W). \tag{41}$$

Assume that all policies are linear-Gaussian: $\pi(a|x) = \mathcal{N}(a|Kx, C)$. In this case, assuming that the policy $\pi$ is stable (the spectral radius of $A + BK$ is less than 1), the stationary state distribution is

$$\mu(x) = \mathcal{N}(0, S), \qquad \text{where } S = (A + BK)S(A + BK)^\top + W. \tag{42}$$

Given an estimate of the dynamics parameters $(\widehat{A}, \widehat{B}, \widehat{W})$, maximum-entropy OPE corresponds to the following convex problem:

$$\max_{S \succeq 0} \ \ln\det(S) \tag{43}$$

$$\text{s.t. } S = (\widehat{A} + \widehat{B}K)S(\widehat{A} + \widehat{B}K)^\top + \widehat{W} \tag{44}$$

Note that the constraint corresponds to that in the dual formulation of LQ control presented in Cohen et al. [2018]. We solve the above problem using cvxpy [Diamond and Boyd, 2016]. The problem will only be feasible if $\rho(\widehat{A} + \widehat{B}K) < 1$, where $\rho(\cdot)$ denotes the spectral radius. Furthermore, when the system is controllable, the constraint fully specifies the solution and so the maximum-entropy objective plays no role.

In LQ control problems, rewards are quadratic:

$$r(x, a) = -x^\top Q x - a^\top R a, \quad Q, R \succ 0, \tag{45}$$

Thus to evaluate policies, we can estimate $Q$ and $R$, and estimate the policy value as

$$\widehat{J}_\pi = \text{trace}(S\hat{Q}) + \text{trace}((KSK^\top + C)\hat{R}).$$

In our experimental setup, we produce the behavior policies by solving for the optimal controller for true dynamics $(A, B, W)$, true action costs $R$, and state costs corrupted as

$$\tilde{Q} = Q + \varepsilon^2 U^\top U$$

$U$ is a matrix of the same size as $Q$ whose entries are generated uniformly at random. Given the corresponding optimal linear feedback matrices $\tilde{K}$, we set behavior policies to $\beta(a|x) = \mathcal{N}(a|\tilde{K}x, 0.1I)$, and we make target policies greedy, i.e. $a = \tilde{K}x$.

When evaluating policies using BRM and FQI, we use the following features for a policy $\pi(a|x) = \mathcal{N}(a|Kx, C)$:

$$\phi(s, a) = \text{VEC}\left(\begin{bmatrix} xx^\top & xa^\top \\ ax^\top & aa^\top \end{bmatrix}\right), \qquad \phi(s, \pi) = \text{VEC}\left(\begin{bmatrix} xx^\top & xx^\top K^\top \\ Kxx^\top & Kxx^\top K^\top + C \end{bmatrix}\right).$$