[Reviews · NeurIPS 2020]

Review 1

Summary and Contributions: This paper describes the use of a maximum-entropy based approach in the training of function approximators (e.g., neural networks) for miniziming the regret in undiscounted, infinite horizon reinforcement learning settings. Theoretical results establish regret bounds on the policy error when the transition and reward functions are linear in features of their inputs, as well as when transition function is approximately linear and the reward function is arbitrary. The approach is evaluated on two synthetic domains and three benchmark problems, highlighting its advantages over existing approaches.

Strengths: The primary strength of the paper is in its theoretical results, which contain proofs in the supplementary material. As far as I could tell, the proofs appeared to be sound. The empirical evaluation also considered multiple domains, including some standard benchmark problems, making it easier to situate the performance across a range of settings. In terms of significance and novelty, the research incrementally builds on recent results (Duan and Wang, 2020) to a broader range of MDPs. The paper is relevant to the reinforcement learning audience within the NeurIPS community.

Weaknesses: The paper is not immediately accessible to an audience without pretty extensive background in linear algebra and an understanding of existing OPE methods in RL. However, given the limited space afforded, I'm not sure what I would recommend trimming in order to provide a more extensive background discussion. It wasn't clear from the presentation and discussion of the empirical results whether and when the approach led to statistically significantly better performance than the baseline approaches (SE bars often overlapped, but it would be 95% CIs we would want to compare to evaluate statistical significance).

Correctness: The method and claims appear correct.

Clarity: The paper is relatively well written, given the extensive amount of very technical content. As discuss above, its accessibility is someone limited.

Relation to Prior Work: The prior literature is appropriately discussed and cited.

Reproducibility: Yes

Additional Feedback: #### Post-Rebuttal Update ##### I thank the authors for their rebuttal. The additional information about the experiments was helpful. Further highlighting in the text that the LQ control is the environment that best evaluates the contributions of the solution and the better performance of the solution would help strengthen the results analysis and reinforce the benefits of the approach.


Review 2

Summary and Contributions: This paper proposes an off-policy evaluation method, following the idea of estimating stationary distribution with function approximation. Under the assumption of linear and ergodic infinite-horizon undiscounted MDP, it is proposed to use maximum entropy principle under feature expectation matching constraints and obtain an exponential family for stationary distribution estimation. In the analysis, the error bound on the learning problem for estimating the stationary distributions is given, followed by the finite-sample error bound of OPE using the learned parameters.

Strengths: The stationary distribution estimation problem stays in the heart of the recent-proposed OPE family, which is a very relevant and important problem for the community. The maximum entropy principle is clean and enjoys convenience in deriving an exponential family of estimators. So this paper makes a nice connection to classic supervised learning. The error bound analysis and the empirical evaluation is sound and clear.

Weaknesses: Section 3.3 policy improvement is a bit of a distraction. The main takeaway is not clear there. In terms of writing, there is room for improvement. For example, a lot of references to the appendix prevent a more smooth flow of the paper. The structure of the paper can also be further optimized. The assumptions in this paper are very strong. In reality, many assumptions would be violated. ========== Thanks for the author response. I appreciate it. For further improvement of the paper, I would recommend actual experiments using neural networks to show generalization of the method to the non-linear feature settings, which may further improve the results. It seems a straightforward extension, since the maxent formulation and the exponential form provide convenience for that.

Correctness: The main technical content seems to be correct. I have the following questions though: When using the linear assumption for the reward and the dynamics, the feature selection/setting is crutial. To relax the linear assumption, it is also mentioned, features can be pre-trained. What would be the recommended way to pre-learn it? For possible violation of the assumptions, how it would affect the results in practice? For example, the logging policy is not estimated accurately or when the single trajectory using the logging policy cannot cover all the states. More discussion would be helpful. In Figure 2, right, the proposed methods (MaxEnt and Model) tend to under-estimate the reward. Is it consistently happening? Any explanation there? The policy improvement is mentioned in the paper and appendix, but never evaluated in the experiments.

Clarity: Here are some suggestions in writing: The related work is after the theory and before the experiments. It would serve the purpose better maybe after experiments. There are several vague statements in the paper, which can be clarified. Like “there are some theoretical advantages...” and “the other methods are better correlated with ...but have somewhat high error”. I would suggest using concrete and clear statements to explain and method and analyze the results. As mentioned, the main takeaway is not clear in 3.3. The analysis there has mixed results and it is generally unclear how to perform policy optimization. So a better option may be just focusing on OPE for the paper and use the space to further illustrate the proposed method or explain the error bound.

Relation to Prior Work: The idea of using maximum entropy principle is also very popular in RL and IRL. I feel a small discussion there would help readers relate different MaxEnt approaches that have been investigated in the general decision making domain.

Reproducibility: Yes

Additional Feedback:


Review 3

Summary and Contributions: This paper studies off-policy evaluation for infinite-horizon undiscounted MDPs with linear function approximation. It provides a finite-sample guarantee of the quality for the estimator. I feel theoretically understanding of OPE estimator in the infinite-horizon undiscounted case is important since recently its importance has been empirically justified: the approach to discounted setting may not work well in an undiscounted setting [1]. I am not aware of any prior work on that, even without linear function approximation. [1]. GenDICE: Generalized Offline Estimation of Stationary Values.

Strengths: This topic is very relevant and beneficial to the NeurIPS community. And as far as I know, there is no prior work that studies the finite-sample property of OPE in an undiscounted setting.

Weaknesses: I feel the experiment part could be improved. If I look at Figure 2, it is a little bit massive to me. The standard deviation is large so some improvements are not statistically significant. From that plot, it may not be easy to get the main point the author wants to deliver. For Theorem 3.2, the authors claim the error scales similarly to the results of Duan and Wang (2020), which nearly match the corresponding lower bound. I am curious this upper bound looks very different with the minimax lower bound in Duan and Wang (2020)? I could not get the point why in the discounting setting, we do not need an entropy regularization (like in Duan and Wang (2020) that is already minimax optimal) but in undiscounted setting, we have to? #################### I have read the rebuttal.

Correctness: I think it is right. In the experiment part, I suggest the author to make the comparison more meaningful. For example, the author comments FQI may lose convergence guarantee and BRM is biased and inconsistent. Those may be better to reflect in the experiment.

Clarity: I think in general it is easy-to-follow and the authors did a great job in reviewing existing works and highlights the pros and cons.

Relation to Prior Work: Very clear! I do appreciate it and it is quite beneficial to the RL community. But they miss one important relevant literature [1] who exactly highlights the necessity of considering the undiscounted setting but without strong finite-sample guarantee. [1]. GenDICE: Generalized Offline Estimation of Stationary Values. ICLR 2020.

Reproducibility: Yes

Additional Feedback:


Review 4

Summary and Contributions: The paper studied the off policy evaluation under undiscounted MDP when the function approximator is linear. Based on the recent work[1], the authors extend the result to the undiscounted case (average reward). Moreover, to get the optimal stationary state distribution, the authors formulate the learning problem as a maximum-entropy problem with a bellman equation constraint. Empirical results show the effectiveness of the proposed method.

Strengths: The authors give a solid analysis of the proposed algorithms, which gives a finite-sample error bound for the average reward case of off-policy evaluation. As far as I know, the result is new.

Weaknesses: A number of related works are missing, for both discussion and comparison. For the undiscounted case, there are a number of works that consider using density ratio to estimate the average reward[2, 3], and it seems that these methods do not require entropy regularization and can also use function approximators to solve the undiscounted MDP case. It would be great for the authors to have a detailed discussion, and a solid empirical comparison with these methods.

Correctness: Yes.

Clarity: Yes.

Relation to Prior Work: Yes. But some parts of the related works are missing, including discussion and empirical comparison.

Reproducibility: No

Additional Feedback: References. [1]. Minimax-optimal off-policy evaluation with linear function approxi- mation [2]Breaking the curse of horizon: Infinite- horizon off-policy estimation [3] GenDICE: Generalized Offline Estimation of Stationary Values ############## I have read the authors' response and other reviewers' comment, and I tend to keep my score.

[Author Response · NeurIPS 2020]

We thank all reviewers for their valuable time and feedback. We believe that we have addressed the main concerns
about the assumptions, related work, and evaluation.

## CONTENT

**R2**, learning the features. The linear MDP assumption $\Pi\Phi = \Phi M$ and $\mathbf{r} = \Phi w$ implies that the transition matrix $\Pi$ is
low-rank and that rewards are linear in the features, so these assumptions can be used to guide feature learning. For
example, for discrete actions we can set $\phi(s,a) = \phi(s) \otimes \delta(a)$, and make $\phi(s|\theta)$ the output of a neural network. The
objective would be to minimize $L(\theta, K, w) := E_{(s,a,r,s')\sim\mathcal{D}} \left[ c_1 \|\phi(s'|\theta) - K\phi(s,a|\theta)\|^2 + c_2(r - \phi(s,a|\theta)^\top w)^2 \right]$.
Note that multiple recent works (offline and online) simply assume a linear MDP with known features in analysis.

**R2**, violation of assumptions. In terms of exploration, we do not require the logging policy to cover all states, but just to
span the features (feature covariance matrix should be full-rank, assumption A3). If this is not the case, the error would
depend on the missing subspace. In terms of linearity, for non-linear MDPs our method would incur an approximation
error. Separating the linearity of dynamics and rewards (rather than implicitly assuming both with a linear Q-function)
at least allows us to drop the linear-reward assumption if we learn the full distribution (simulated in Figure 1, middle).

**R3, R4**, necessity of entropy. Our finite-sample guarantees for $J_\pi$ actually hold for any distribution satisfying the
constraint (see Remark 1) as long as the MDP is linear, and we can also use the simple estimate in Equation (10).
Empirically, we find entropy to be a good regularizer when learning the full distribution $d_\pi(s,a) = \mu_\pi(s) \otimes \pi(a|s)$.
Learning $d_\pi$ is useful in the case of non-linear rewards, or $E_d[r(s,a)]$ cannot be computed in closed form. Appendix D
gives a justification of the maximum-entropy objective for MDPs with sufficiently random dynamics. More generally,
maximizing entropy is equivalent to minimizing KL-divergence to the uniform distribution. We can use the same
KL-divergence formulation to impose different distribution priors when available.

**R3**, comparison to bounds in Duan and Wang (2020). These bounds are expressed in terms of $\chi^2$ divergence, and for
linear $f$, they are a function of the spectrum of the feature covariance matrix $\Sigma$. This also the case with our bound - it
scales inversely with the smallest eigenvalue of $\Sigma$. The results are admittedly different otherwise (we will clarify this in
the paper). Note also that the bounds of Duan and Wang (2020) scale with the horizon / effective horizon, and are thus
infinite in our setting. Furthermore, their discounted infinite-horizon bound scales as $N^{-1/2}$ where $N$ is the number of
trajectories, whereas our bound scales as $T^{-1/2}$ where $T$ is the number of transitions (possibly in a single trajectory).

## EXPERIMENTS

**R1, R3**, we agree the the methods are not always well-separated in the sense of 95% confidence intervals, and BRM and
FQI can perform well. However, note that some of the evaluated environments violate our assumptions (Taxi, Acrobot),
and on these our method performs at least as well as the baselines. The only evaluated environment satisfying linearity
and ergodicity is LQ control (Figure 2 left), where our approach is clearly better. Concretely, the model-based method
is clearly not sensitive to $\varepsilon_1$ (controlling level of suboptimality of the target policy), while BRM and FQI deteriorate
with larger $\varepsilon_1$, and the curves are well-separated for $\varepsilon_1 \geq 0.2$.

**R2**, for Acrobot, it is an interesting observation that MaxEnt and Model often underestimate the expected reward
(actually so does FQI). However, for smaller true $J_\pi$, the same methods overestimate. This may be an artifact of
covariance regularization for linear regression in the model / Q-function.

**R3**, we chose not to explicitly demonstrate divergence of FQI and biasednesss of BRM since these issues are well
known (see references in Section 4).

**STYLE**. **R2**, thank you very much for the suggestions. We agree about Section 3.3 and in retrospect should have saved
a discussion of policy improvement for future work. We will remove it and move some of the Appendix into the paper.

## RELATED WORK

**R2**, thank you for the suggestions. We will add references to maximum-entropy approaches in RL and IRL. In RL,
entropy is typically a regularizer on the policy, rather than the state-action distribution. IRL approaches are more similar
to ours, maximizing the entropy of distribution over paths s.t. feature expectations match demonstration data.

**R4**, thank you, we will improve the discussion. We actually do refer to the "Breaking the Curse of the Horizon" work
(lines 24,65, 204-208). Their formulation also relies on the relaxed feature expectation constraint. They minimize the
constraint violation while maximizing over features, and have a consistency guarantee. We assume oracle features and a
linear MDP, satisfy an approximate constraint (given by the model estimate), and show finite-sample guarantees.

**R3, R4**. We will add a reference to GenDICE. In the paper, we refer to "RL via Fenchel-Rockafellar Duality" by
Nachum and Dai (2020), which provides a unified view of the DICE papers, including GenDICE. Section 7 there
discusses the undiscounted setting via different DICE methods. We compare to the basic Lagrangian formulation in our
paper (lines 195-205), and will add a note on GenDICE (regularized Lagrangian).

[Meta-Review · NeurIPS 2020]

This is a borderline paper. The paper is technically sound and addressing OPE in average-reward setting is an important problem. Despite that the work is an extension of Duan and Wang (for discounted setting) to the average-reward setting, the algorithm is somewhat different, as Duan and Wang uses FQE whereas the current paper performs stationary-distribution estimation. That said, there are a few weaknesses that the paper should try to address or at least discuss: 1. The entropy maximization is a novel algorithmic element which does not appear in previous approaches in the discounted setting. Naturally multiple reviewers questioned the necessity of the approach. In the rebuttal, the authors clarified that this helps with empirical performance and is justified with additional assumptions, but does not seem necessary for the core theory. Since this paper is positioned mainly as a theory paper, this should be carefully clarified so that theory readers can have the correct takeaway messages from the paper without being confused, and possibly de-emphasize the role of max-ent in the paper (or even in the title). 2. Strength of result: While the paper (L33) mentions that the bound scales similarly to that of Duan and Wang, the reviewers have noted in the discussion that the results are comparatively weaker and require stronger assumptions: the bound is not adaptive to how close the target and behavior policies are, and requires the additional assumption that target policy also induces an exploratory distribution (L118). Is it a weakness of the analysis, or is it fundamentally difficult to prove such a result in the average-reward setting? 3. The estimation procedure in Eq.11 and 12 is highly similar to LSTD, which is generalized by many recent methods that approach OPE from LP and duality. So no wonder the authors find similarity with DualDICE in L196, as many of these recent methods collapse to some form of LSTD in the linear case [1,2]. Moreover, the authors should also discuss existing finite-sample analysis of LSTD family; see e.g., [3,4]. [1] Uehara et al'20. Minimax Weight and Q-Function Learning for Off-Policy Evaluation. [2] Yang et al'20. Off-Policy Evaluation via the Regularized Lagrangian. [3] Lazaric et al'10. Finite-sample analysis of LSTD. [4] Liu et al'15. Finite-Sample Analysis of Proximal Gradient TD Algorithms.